# Nitrogen-Doped Porous Carbon Derived from Coal for High-Performance Dual-Carbon Lithium-Ion Capacitors

**DOI:** 10.3390/nano13182525

**Published:** 2023-09-09

**Authors:** Jiangmin Jiang, Qianqian Shen, Ziyu Chen, Shijing Wang

**Affiliations:** 1Jiangsu Province Engineering Laboratory of High Efficient Energy Storage Technology and Equipments, School of Materials Science and Physics, China University of Mining and Technology, Xuzhou 221116, China; 2School of Materials Science and Engineering, Zhejiang University, Hangzhou 310027, China; 3Tsinghua Shenzhen International Graduate School, Tsinghua University, Shenzhen 518055, China

**Keywords:** lithium-ion capacitors, dual-carbon lithium-ion capacitors, coal-based carbon materials, nitrogen doping, pore structure regulation

## Abstract

Lithium-ion capacitors (LICs) are emerging as one of the most advanced hybrid energy storage devices, however, their development is limited by the imbalance of the dynamics and capacity between the anode and cathode electrodes. Herein, anthracite was proposed as the raw material to prepare coal-based, nitrogen-doped porous carbon materials (CNPCs), together with being employed as a cathode and anode used for dual-carbon lithium-ion capacitors (DC-LICs). The prepared CNPCs exhibited a folded carbon nanosheet structure and the pores could be well regulated by changing the additional amount of g-C_3_N_4_, showing a high conductivity, abundant heteroatoms, and a large specific surface area. As expected, the optimized CNPCs (CTK-1.0) delivered a superior lithium storage capacity, which exhibited a high specific capacity of 750 mAh g^−1^ and maintained an excellent capacity retention rate of 97% after 800 cycles. Furthermore, DC-LICs (CTK-1.0//CTK-1.0) were assembled using the CTK-1.0 as both cathode and anode electrodes to match well in terms of internal kinetics and capacity simultaneously, which displayed a maximum energy density of 137.6 Wh kg^−1^ and a protracted lifetime of 3000 cycles. This work demonstrates the great potential of coal-based carbon materials for electrochemical energy storage devices and also provides a new way for the high value-added utilization of coal materials.

## 1. Introduction

Fossil energy occupies a dominant position in the current world energy landscape, whereas, due to non-renewable and environmental pollution shortcomings, it cannot meet long-term use by human beings. Therefore, it is necessary to find new green energy to replace part of fossil energy [1,2,3,4,5]. Electrochemical energy storage can provide a powerful energy storage system to achieve stable and continuous outputs [6,7,8]; thus, it has become one of the hot spots in the new century [9]. Among them, supercapacitors (SCs) have an excellent cycle life (>10^6^ cycles) and high power density (>10 kW kg^−1^), but are restricted by a low energy density. On the contrary, lithium-ion batteries (LIBs) can provide prominent energy density (>250 Wh kg^−1^), but the slow Li^+^ diffusion process in the electrode results in a low power density. To achieve high energy and power density, a good long cycle life, and a low cost for energy storage devices, a promising approach is to assemble hybrid device lithium-ion capacitors (LICs) [10,11,12,13,14,15]. LICs are made of lithium intercalation/de-intercalation-type anode material, together with capacitive-type cathode material with fast kinetics [16,17,18]. This combination can achieve a superior energy performance, bridging the energy density and power density performance gap between LIBs and SCs [19,20,21,22].

Recently, various electrode materials of LICs have been widely investigated. Capacitor-type cathode materials mainly include carbonaceous materials [23], such as activated carbon [24], template carbon [25], graphene [26], and graphene oxide. Battery-type anode materials mainly include transition metal oxides [25], carbonaceous materials [27], Mxene [28], SiOx [29], Li_4_Ti_5_O_12_ [30], and carbon composites [31]. Among them, carbon-based materials with an incredibly high chemical stability, excellent electrical conductivity, and abundant and inexpensive advantages are promising candidates for electrode materials. LICs assembled with carbon-based materials exhibit excellent cycling capabilities and fast charging performances. However, the non-faraday reactions of the cathode limit the specific capacity of LICs, and the slow Li^+^ insertion dynamics of the anode limit the intercalation/de-intercalation of the device. In this regard, LICs face the bottleneck of an imbalance between the cathode and anode in the kinetics process, as well as the cycle life still being insufficient up to now.

To solve the issue of the limited performance of LICs, scholars have conducted a lot of research works on the development of novel materials and the structural design of hybrid capacitors. In particular, the use of carbon-based materials, such as cathode and anode materials simultaneously, to assemble dual-carbon lithium-ion capacitors (DC-LICs) is an emerging means for solving the imbalance of dynamics and capacity. Noting that DC-LICs are assembled with the same material as the anode and cathode electrodes, which can use almost the same material precursors and the same preparation technologies [32,33,34], this leads to greatly simplifying the process flow and having great application prospects. For this purpose, some kinds of DC-LICs have been proposed, such as N-doped graphene (NG//NG) [35], N-doped carbon nanospheres (NCS//NCS) [36], and B, N-doped carbon nanofibers (BNC//BNC) [37], etc. The research results show that the lithium storage performance of DC-LICs has been greatly improved compared to that of traditional LICs. However, most sources of carbon used for electrode materials are not cheap and, in addition, capacitor-type cathode electrodes exhibit a low capacity and battery-type anodes show slow electrochemical kinetics, leading to the overall electrochemical performance of DC-LICs not being ideal. Thus, the development of low-cost and high-performance DC-LICs has been challenging so far.

In this work, coal-based, nitrogen-doped porous carbon materials (CNPCs) were synthesized through a facile one-step carbonization method. We noted g-C_3_N_4_ with a high nitrogen content and layered structure as the template agent and nitrogen source, which exerted a vital role in regulating the pore structure and heteroatom doping. When the additional amount of g-C_3_N_4_ was 1.0 g, the optimized CNPCs of CTK-1.0 exhibited a folded carbon nanosheets structure and large specific surface area, which could be used as a carbon-based anode with improved lithium ion insertion kinetics and a highly porous carbon-based cathode with an enhanced ion adsorption capacity. As a result, DC-LICs (CTK-1.0//CTK-1.0) were successfully assembled using the CTK-1.0 sample as both a cathode and anode, which exhibited a maximum energy density of 137.6 Wh kg^−1^ and a protracted lifetime of 3000 cycles, showing promising application prospects.

## 2. Materials and Methods

### 2.1. Synthesis of Coal-Based Porous Carbon Materials

All the reagents were of analytical grade and were used without any further purification. Based on a typical synthesis method and modified, the g-C_3_N_4_ template was synthesized via direct pyrolysis at 500 °C at 15 °C min^−1^ for 3 h. In a typical procedure, 1.0 g of anthracite, 4.0 g of K_2_C_2_O_4_·H_2_O, and varying addition amounts of g-C_3_N_4_ (0.6, 1.0, and 1.4 g) are evenly mixed in a ball mill. The fully mixed powder was then put into a tubular furnace and pyrolyzed at 800 °C for 2 h in a flowing Ar atmosphere at a heating rate of 3 °C min^−1^ and then cooled to room temperature. Finally, the resulting black powders were pickled with hydrochloric acid to remove impurities, then washed with deionized water to a pH of 7, and further dried overnight at 80 °C. When the addition amounts of g-C_3_N_4_ were 0.6, 1.0, and 1.4 g, the obtained coal-based, nitrogen-doped porous carbon materials (CNPCs) were named CTK-0.6, CTK-1.0, and CTK-1.4, respectively. For comparison, the anthracite with no template and activator was heated under the same conditions, and the obtained sample was named CT.

### 2.2. Material Characterization

The morphology and microstructure were observed using a 5 kV field emission scanning electron microscope (FESEM, SU82205). A transmission electron microscope (TEM) and FEI Tecnai 20 high resolution (HRTEM) were used. XRD patterns were obtained on a powder X-ray diffractometer (Miniflex, Rigaku) using Cu-Kα radiation (λ = 1.5406 Å). An X-ray photoelectron spectrometer (ESCALAB 250Xi, with Al Kα X-ray source) was used to measure the elemental composition of the sample, and all the spectra were corrected with reference to C 1s at 284.8 eV. The laser confocal Raman spectrometer of Senterra was used to analyze the properties of the CNPCs. N_2_ adsorption–desorption measurements were performed using a surface area analyzer (ASAP 24603.01.02) to determine the specific surface area and pore volume, and the specific pore size distribution was determined using density functional theory (DFT).

### 2.3. Electrochemical Measurement

The preparation of the half cell: in order to prepare the anode, a slurry containing 70 wt% active material (CNPCs), 20 wt% carbon black, and 10 wt% sodium carboxymethyl cellulose (CMC) was ground evenly. Distilled water was used as the solvent to stir fully until a paste was obtained, and then the slurry was coated to the surface of copper foil. After drying at 60 °C for 12 h in a vacuum drying oven, the anodes were cut into suitable sizes. Lithium metal was used as the reference electrode and counter electrode, and the active material was used as the working electrode. Polypropylene microporous membrane was used as the separator, and 1 M LiPF_6_ dissolved in a mixture of ethyl carbonate (EC)/diethyl carbonate (DEC)/dimethyl carbonate (DMC) was used as an electrolyte for button cell assembly in a glove box filled with argon.

The preparation of the DC-LICs: the cathode was prepared in the same manner as the anode, except using aluminum foil as the current collector. Before the assembly of the DC-LICs, the anode was pre-lithiated to eliminate the initial irreversible capacity loss. The assembly of the DC-LICs was carried out in the same assembly way as the button cell assembly, where the same organic electrolyte was used for the pre-lithium anode and cathode, and the mass ratio of anode: cathode was 1:1.

A CT20001A battery testing instrument (LAND Electronic Co., Ltd, Wuhan, China.) was used to perform constant current charge/discharge tests at different current densities. Cyclic voltammetry (CV) and electrochemical impedance spectroscopy (EIS) were performed using the CHI 760E electrochemical workstation. The energy density (*E*, Wh kg^−1^), specific capacitance (*C*, F g^−1^), and power density (*P*, W kg^−1^), based on the total mass of the two electrode materials, can be calculated from the following equation:$$\begin{array}{c} E=\int_{t_{1}}^{t_{2}}\;IVdt=\frac{1}{2}C\left(V_{max}+V_{min}\right)\left(V_{max}-V_{min}\right)\\ C=\frac{I\times t}{m\times \Delta V}\\ P=\frac{E}{t} \end{array}$$
where $V_{max}$ and $V_{min}$ are the voltage values at the end and beginning of the charge curve during the constant current charging/discharging process, *I* is the current density, ΔV is the difference between $V_{max}$ and $V_{min}$, *m* is the total mass of the active material in the two electrodes, and t is the discharge time.

## 3. Results and Discussion

Graphite-phase carbon nitride (g-C_3_N_4_) is a typical organic polymer that is composed of a large number of carbon and nitrogen elements. The two atoms form a large π bond through sp^2^ hybridization and become a highly delocalized conjugated system. The structure is usually made up of triazine rings, or 3-s triazine rings, which are graphite-like layers connected by weak van der Waals forces [38]. Due to its properties of a high nitrogen content and easy decomposition, g-C_3_N_4_ could be used as both the template agent and nitrogen source in this experiment. According to the scheme shown in Figure 1a, the anthracite, g-C_3_N_4_, and K_2_C_2_O_4_·H_2_O powders were heated in an inert atmosphere. The g-C_3_N_4_ template was decomposed when the temperature was higher than 600 °C (Appendix A), and various nitrogen-containing gases ($C_{2}N_{2}^{+}$, $C_{3}N_{2}^{+}$, and $C_{3}N_{3}^{+}$) produced by decomposition would lead to the in situ pore-making and nitrogen-doping of the coal-based carbon materials, thus forming the final products of the CNPCs.

To investigate the effect of the addition of g-C_3_N_4_ to the microstructure, the prepared CNPCs were tested with SEM and TEM. Compared to CT, the CTK-0.6, CTK-1.0, and CTK-1.4 samples showed a rougher interface (Figure 1b–e). This might have been related to the fact that the gas produced in the activation process made pores in the carbon materials. The loose pore structure of the carbon material could enhance the specific surface area of the materials and accelerate the diffusion of the electrolyte, thus speeding up the Li^+^ transmission rate [39]. As shown in Figure 1f, the CT sample displayed a thicker structure, the surface was smooth, and hardly any wrinkles could be observed, while the CTK-0.6, CTK-1.0, and CTK-1.4 samples exhibited a thin, two-dimensional geometry with a wrinkled or corrugated structure (Figure 1g–i). The formation of folded carbon nanosheets was conducive to accelerating the transfer of electrolyte ions [39]; it might have been caused by the high internal pressure blowing away the carbon matrix of a large amount of gas released by the g-C_3_N_4_ during high-temperature pyrolysis. Additionally, the graphene-like nanosheets became thinner with the increase in the additional amount of the g-C_3_N_4_.

The HR-TEM image of CTK-1.0 is shown in Figure 1j. It can be seen that there were many carbon defects along the lattice fringe originating from dislocations or discontinuous lattices (yellow circles), which can provide more active sites for lithium storage [40]. The calculated carbon layer spacing of CTK-1.0 was 0.477 nm, which may have been due to the increase in the carbon layer distance caused by the nitrogen doping; this was conducive to reducing the energy barrier of the Li^+^ intercalation/de-intercalation and improving the capacity. As shown in the STEM and energy dispersive X-ray energy spectra (EDS) of CTK-1.0, the C, N, O, and S elements existed in the carbon skeleton of the CNPCs (Figure 1k), in which the source of sulfur was inherent to the precursor of coal. It can be seen that the C and N elements were uniformly distributed in the CNPCs (Figure 1l,m), indicating the successful introduction of heteroatomic nitrogen.

As shown in Appendix A, the XRD pattern of g-C_3_N_4_ has two peaks at 12.9° and 27.5°, which were (100) and (002) crystal surfaces, respectively [41,42,43]. In the XRD patterns of the prepared CNPCs (Figure 2a), the peaks of g-C_3_N_4_ disappeared, and this further confirmed that there was no residue of g-C_3_N_4_ in the obtained CNPCs. In particular, the XRD patterns of CTK-0.6, CTK-1.0, and CTK-1.4 exhibited two diffraction peaks at around 25.9° and 43.9°, which corresponded to the (002) and (100) planes of graphite, respectively. It should be noted that the two diffraction peaks were diffuse and weak, indicating that the prepared CNPCs were an amorphous structure. Compared to CT, it can be seen that the (002) peak of the CTK-0.6, CTK-1.0, and CTK-1.4 samples was slightly shifted to the left. According to the Bragg Equation 2dsinθ=nλ, the surface spacing of the graphite microcrystals of the CNPCs increased slightly after nitrogen doping [40], which might have contributed to reducing the energy barrier for Li^+^ in the intercalation and de-intercalation processes.

Raman spectra were used to evaluate the degree of defects and disorder of the prepared carbon materials. As shown in Figure 2b, the prepared CNPCs showed wide 2D peaks, which confirmed the relatively high degree of graphitization and presence of multiple graphene layers to some extent. There were two typical peaks of the carbon-based materials at about 1342 and 1585 cm^−1^, which corresponded to defect-induced (D band) and graphite-induced carbon structures (G band). The ratio of peak D intensity to peak G intensity (I_D_/I_G_) can be used to evaluate the degree of defects and disorder in carbon materials [44]. In particular, the I_D_/I_G_ of CT, CTK-0.6, CTK-1.0, and CTK-1.4 were 0.91, 0.95, 0.97, and 0.98, respectively. This demonstrated that, with an increase in the g-C_3_N_4_ content, the I_D_/I_G_ value gradually increased, indicating that the addition of g-C_3_N_4_ led to more structural defects and active sites.

The chemical states and element compositions of the CNPCs were investigated using XPS technology. The full XPS spectra of the CNPCs showed the presence of C, N, and O elements (Figure 2c), and the detailed surface atomic contents are summarized in Appendix A. It can be found that, with the additional amount of g-C_3_N_4_ increases, the N atomic percentage increased from 1.26 at.% (CTK-0.6) to 3.03 at.% (CTK-1.0) and finally to 4.34 at.% (CTK-1.4). The high-resolution C 1s spectra of the CTK-1.0 sample are shown in Figure 2d, and three different characteristic peaks were located at 284.8 eV, 286.1 eV, and 289.5 eV, representing the C-C, C-O, and O-C=O bonds, respectively [45]. The high-resolution XPS spectra of O 1s could be fitted into three characteristic peaks (Figure 2e), which were located at 530.6 eV, 532.1 eV, and 533.3 eV, corresponding to the C=O, C-O-C, and O-C=O bonds, respectively. It should be noted that O atoms mainly come from water or oxygen molecules and are closely related to pore structure [46,47]. The high-resolution XPS spectra of N 1s could be divided into three different peaks (Figure 2f), which corresponded to the pyridine-N (398.1 eV), pyrrole-N (400.4 eV), and quaternary-N (402.7 eV), respectively. Among them, pyridine-N and pyrrole-N can serve as faraday reaction sites and provide pseudo-capacitance contributing to facilitating the fast charging performance, together with the quaternary-N, which can promote electron transfer in the carbon lattice, thus improving the electronic conductivity of carbon-based materials [48].

To further explore the pore structure of the prepared CNPCs, N_2_ adsorption–desorption isotherm was carried out to study the porous structure. As shown in Figure 2g, CTK-0.6, CTK-1.0, and CTK-1.4 presented the typical adsorption–desorption isotherms of type I [31], indicating that the CNPCs exhibited a large amount of microporous structures. It should be noted that the internal reason for this was that K_2_CO_3_ crystals can be formed by the heat treatment of potassium oxalate, which further decomposes to produce CO_2_ and K_2_O, CO_2_ to further etch carbon to produce micropores at a higher temperature [49]. In addition, the corresponding pore size distribution indeed confirmed the microporous layered structure; the specific surface areas of CTK-0.6, CTK-1.0, and CTK-1.4 were 1479.3 m^2^ g^−1^, 1673.5 m^2^ g^−1^, and 848.8 m^2^ g^−1^, and the pore volumes were 0.61 cm^3^ g^−1^, 0.77 cm^3^ g^−1^, and 0.39 cm^3^ g^−1^, respectively, (Appendix A). With the change in g-C_3_N_4_ content, although all the CNPCs could obtain pore structures with micropores, the pore size distributions of the micropores were different (Figure 2h). It can be seen that the specific surface area and pore volume of CTK-1.0 were higher than those of CTK-0.6 and CTK-1.4. Notably, this unique multi-scale porous structure of active surfaces with a high specific surface area can infiltrate more electrolytes, shorten diffusion paths, and minimize high-rate diffusion losses, which facilitates electrochemical reaction kinetics.

The electrical performances of the CNPCs were measured within the potential range of 0.01~3.0 V (vs. Li/Li^+^). In the first three cyclic voltammetry (CV) curves of the CTK-1.0 electrode at 0.1 mV s^−1^, two reduction peaks can be observed near 0.4 V and 0.01 V in the initial cycle (Figure 3a and Appendix A), corresponding to the formation process of the solid electrolyte interface (SEI) and the process of Li^+^ embedding into the graphite layer, respectively [48]. Notably, the peak disappeared in the subsequent charge–discharge process, and the CV curves of the second and third cycles were highly coincident, indicating that the Li^+^ intercalation/de-intercalation processes of the CTK-1.0 had a good reversibility (Appendix A). The constant charge/discharge curves of the CTK-1.0 sample were tested at 0.1 A g^−1^, and the long platform at 0.01~0.8 V was related to the intercalation of Li^+^ (Figure 3b). CTK-1.0 delivered a high reversible specific capacity of 750 mAh g^−1^, and the irreversible capacity observed from the initial charge–discharge curves and CV curves was also mainly scribed to the SEI formation and decomposition of the electrolyte. The subsequent curves with nearly overlapping cycles, which verifies the excellent stability of the SEI layer.

The rate performance of the CNPCs under different addition amounts of g-C_3_N_4_ is shown in Figure 3c. With the current density increased from 0.1 to 5.0 A g^−1^, the reversible capacities of CTK-0.6, CTK-1.0, and CTK-1.4 decreased from 400, 750, and 550 mAh g^−1^ at 0.1 A g^−1^ to 70, 167, and 150 mAh g^−1^, respectively. When the current density was restored to 100 mA g^−1^ after the cycles, their specific capacities rose to 300, 560, and 450 mAh g^−1^, respectively. It can be seen from the specific capacity and current density images in Figure 3d that the specific capacities of the CNPCs were negatively correlated with the current densities. The diffusion rate of the lithium-ion inside the button cell could not meet the requirements of the high current when the current densities were increased, resulting in the intensification of electrochemical polarization [50]. Therefore, the specific capacity of the CNPCs increased first and then decreased when increasing the addition of g-C_3_N_4_, and the optimized CTK-1.0 exhibited the best rate performance.

To further study the cyclic stability of the prepared CNPCs, high current and long period cycle tests were carried out (Figure 3e and Appendix A). Before the charge–discharge test, a low current density of 0.1 A g^−1^ was used for five cycles to promote the formation of a dense and stable SEI film on the surface of the CNPCs. After that, a high current density of 2.0 A g^−1^ was applied to test their cycling performance. As shown in Figure 3e, with the deepening of the cycle, the Coulombic efficiencies of CT, CTK-0.6, CTK-1.0, and CTK-1.4 were all higher than 99%. After 800 cycles, the reversible capacities of the CT, CTK-0.6, CTK-1.0, and CTK-1.4 electrodes decreased from 225.5, 146.4, 315.6, and 219.6 mAh g^−1^ to 155.8, 129.4, 238.9, and 129.4 mAh g^−1^, respectively. Compared to the other three groups of samples, the capacity retention rate of optimized CTK-1.0 was up to 97%, showing the best cyclic stability. This might have been due to the appropriate defects and stable degree of the heteroatom doping on the surface of the CNPCs during the activation process, which provided more active sites for Li^+^ storage [51], leading to a better rate performance and cycle stability.

To clarify the specific Li^+^ storage mechanism, the CV curves of the CNPCs were tested at different scanning rates (Appendix A). As shown in Figure 4a, when these scanning rates increased from 0.1 mV s^−1^ to 2.0 mV s^−1^, the enclosed area of the CV curves gradually increased, and the shape of the redox peaks did not change significantly, but their positions moved to low potential and high potential, respectively [52]. The specific capacitive mechanism can be decided by the following equation:$$i={av}^{b}$$
where *i* is the peak current, v is the scanning rate, and *a* and *b* are adjustable constants. By linear fitting the log (*i*)-log (v) curve, the value of *b* can be calculated. When the *b*-value approaches 0.5, the electrochemical reaction is dominated by a diffusion-controlled intercalation process, whereas a *b*-value near 1.0 indicates a capacitive-controlled process [51]. As shown in Figure 4b, the CTK-1.0 electrode had a b value approximately equal to 0.7 within scanning rate range of 0.1–2.0 mV s^−1^, indicating the coexistence of capacitive behavior and diffusion control behavior. The synergistic effect of the capacitive behavior and diffusion behavior of the material was conducive to realizing rapid charge transfer and an enhanced long-term cycling stability [51]. In addition, the capacitive contribution of the Li^+^ storage to the total capacity can be calculated based on the following equation [53]:$$I(V)=k_{1}v+k_{2}v^{1/2}$$
where $k_{1}$ and $k_{2}$ are constants by linear fitting, with $k_{1}v$ and $k_{2}v^{1/2}$ corresponding to diffusion-controlled and pseudocapacitive-controlled processes, respectively. The contribution of the capacitance and diffusion behavior of CTK-1.0 at different scanning rates is shown in Figure 4c; as the scanning rate increased from 0.1 to 2.0 mV s^−1^, the pseudocapacitive-controlled behavior contribution ratio for CTK-1.0 improved from 28% to 63%. It can be seen that an increase in the scanning rate would lead to an increase in the contribution rate of the capacitive lithium storage of CTK-1.0, indicating that capacitive behavior played a key role in the high scanning rate [54]. The prepared CTK-1.0 electrode material contained abundant pores, surface microscopic defects, and nitrogen atoms, which would greatly increase the surface adsorption site of Li^+^, thereby increasing the capacitive surface adsorption behavior.

To better understand the influence of g-C_3_N_4_ on the electrochemical behavior of the materials, the galvanostatic intermittent titration technique (GITT) was performed on the prepared electrodes (tested at 0.1 A g^−1^, relaxation time 30 min, and pulse time 10 min). As displayed in Figure 4d, the apparent diffusion coefficients of lithium-ion (*D*_Li+_) in different electrodes can be calculated based on the following equation [55]:$$D_{\mathrm{Li}+}=\frac{4}{\pi \tau }{\left(\frac{m_{B}V_{M}}{M_{B}S}\right)}^{2}{\left(\frac{∆E_{S}}{∆E_{t}}\right)}^{2}$$ where $\;m_{B}$ is the mass of the active substance (g), $M_{B}$ is molar mass (g mol^−1^), $V_{M}$ is the molar volume, S is the area of the electrode (cm^2^), τ is the relaxation time (s), $∆E_{S}$ is the change in the steady-state potential through the current pulse, and $∆E_{t}$ is the change in potential after the current pulse minus iR drop. *D*_Li+_ in the discharge/charge process can be calculated according to the GITT curves. As shown in Figure 4e,f, the *D*_Li+_ of CT was centered at around 3.0 × 10^−10^~2 × 10^−9^ cm^2^ S^−1^, and the *D*_Li+_ of CTK-1.0 was concentrated in the vicinity of 1.47 × 10^−10^~2.7 × 10^−9^ cm^2^ S^−1^. The diffusion coefficients of the CT and CTK-1.0 electrodes changed similarly in the process of Li^+^ insertion and disinsertion. In the process of lithium intercalation (Figure 4e), the *D*_Li+_ first decreased slowly with a decrease in voltage, and then decreased rapidly from about 0.5 V. In the process of Li^+^ de-intercalation (Figure 4f), the *D*_Li+_ decreased slowly with an increase in voltage, and then increased when it rose to 2.0 V. In particular, the general diffusion coefficient trend of CTK-1.0 was higher than that of CT, demonstrating that the lithium diffusion rate in the prepared CTK-1.0 material was faster and the dynamic performance was better than that in the comparison CT samples.

Electrochemical impedance spectroscopy (EIS) is also a reliable method for analyzing reaction kinetic behavior. Nyquist plots of different samples when the discharge potential was controlled at 0.7 V are shown in Figure 4g. The specific resistances of the electrode materials can be estimated from the circular radius of the high-frequency region in the Nyquist plot [56]. The semicircle diameter of CTK-1.0 was the smallest, implying a minimal R_ct_ among the CNPC materials, which meant that the conductivity and ion transfer rate between the electrode material and electrolyte were better. The low transfer resistance for CTK-1.0 might be attributed to the large carbon interlayer distance and N-doped structure, which enhanced the charge transfer process significantly [57]. Additionally, with an increase in the charging voltage, the resistance of the electrode material of CTK-1.0 showed an increasing trend (Figure 4h); this phenomenon is common behavior of anode electrode materials.

To investigate their practical application potential, the DC-LICs (CTK-1.0//CTK-1.0) were assembled and evaluated in Figure 5. It should be noted that the DC-LICs were fabricated using CTK-1.0 as both the cathode and the anode electrodes. As shown in Figure 5a, the DC-LICs showed a typical hybrid mechanism, which combined the battery storage mechanism and capacitor energy storage mechanism simultaneously. During the charging process, the anions in the electrolyte migrated to the cathode electrode, and Li^+^ was embedded in the anode electrode, while the discharge was the opposite process [58]. Before the device assembly, the CTK-1.0 used as the anode was pre-lithium for 1 h by an internal short circuit method to eliminate the initial irreversible reaction [59]. The assembled DC-LICs could operate in a wide voltage window of 0.1–4.2 V. It can be seen that there were similar-shape characteristics of the CV curves of the DC-LICs at different scanning rates from 1.0 mV s^−1^ to 100 mV s^−1^ (Figure 5b), indicating a good reversible process.

The constant current charge–discharge curves of the assembled DC-LICs under different current densities presented an approximate linear slope without an apparent potential platform (Figure 5c); this further confirmed the typical hybrid storage mechanism. In particular, the calculated specific capacitance values of the DC-LICs were 56.1, 50.8, 41.2, 38.9, 33.2, and 29.3 F g^−1^ (based on the total mass of the cathode and anode active materials) at the current densities of 0.1, 0.2, 0.5, 1, 2, and 5 A g^−1^, respectively. On this basis, the energy density and power density of the assembled DC-LICs can be calculated, and, under the power density of 410 W kg^−1^, the energy density of the DC-LICs was as high as 137.62 Wh kg^−1^. Even under a high power density of 20.5 kW kg^−1^, the energy density could still reach 71.8 Wh kg^−1^ (Figure 5d). In addition, our designed DC-LICs (CTK-1.0//CTK-1.0) had better performances compared to the other hybrid capacitors in terms of energy/power characteristics, such as graphitic carbon nanosheets//activated carbon [60], titanium carbide//activated carbon [61], graphitic carbon//sisal fiber activated carbon [62], Li_4_Ti_5_O_12_-graphene//activated carbon [63], T-Nb_2_O_5_@C//commercial activated carbon [64], and commercialized lithium-ion capacitors et al. (Appendix A). Moreover, the cycling performances of the assembled DC-LICs were investigated (Figure 5e). They exhibited a superior cyclic stability, which displayed a high capacity ratio of 75% at a current density of 2 A g^−1^ after 3000 cycles, showing broad prospects for practical applications.

## 4. Conclusions

In summary, coal-based, nitrogen-doped porous carbon (CNPCs) with a folded carbon nanosheets structure were proposed using a simple one-step carbonization method, whose structures and heteroatoms doping could be well regulated by changing the proportion of g-C_3_N_4_. The interconnected and porous carbon nanosheets provided a large specific surface area in contact with the electrolyte, and the nitrogen-doping structure offered additional active sites for pseudocapacitance and Li^+^ adsorption. As a result, the optimized CTK-1.0 exhibited an excellent lithium storage capacity, which delivered a high specific capacity of 750 mA g^−1^ and maintained a capacity retention rate of up to 97% after 800 cycles. Furthermore, DC-LICs (CTK-1.0//CTK-1.0) were assembled using CTK-1.0 as a cathode and anode simultaneously, which exhibited a maximum energy density of 137.6 Wh kg^−1^, together with a superior lifetime of up to 3000 cycles. Significantly, this work opens up a new path for low-cost and advanced carbon materials for the practical applications of next-generation energy storage technologies.

## Figures and Tables

**Figure 1 nanomaterials-13-02525-f001:**
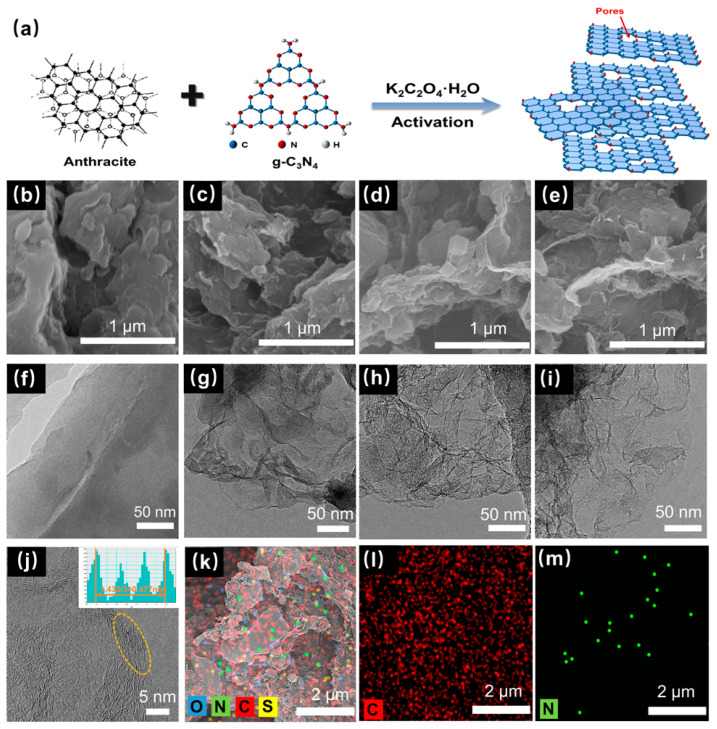
(**a**) Schematic diagram of the synthesis scheme of CNPCs. SEM image of (**b**) CT, (**c**) CTK-0.6, (**d**) CTK-1.0, and (**e**) CTK-1.4. TEM images of (**f**) CT, (**g**) CTK-0.6, (**h**) CTK-1.0, and (**i**) CTK-1.4. (**j**) HR-TEM and (**k**) STEM images of CTK-1.0 with the corresponding (**l**) C (red) and (**m**) N elemental mappings.

**Figure 2 nanomaterials-13-02525-f002:**
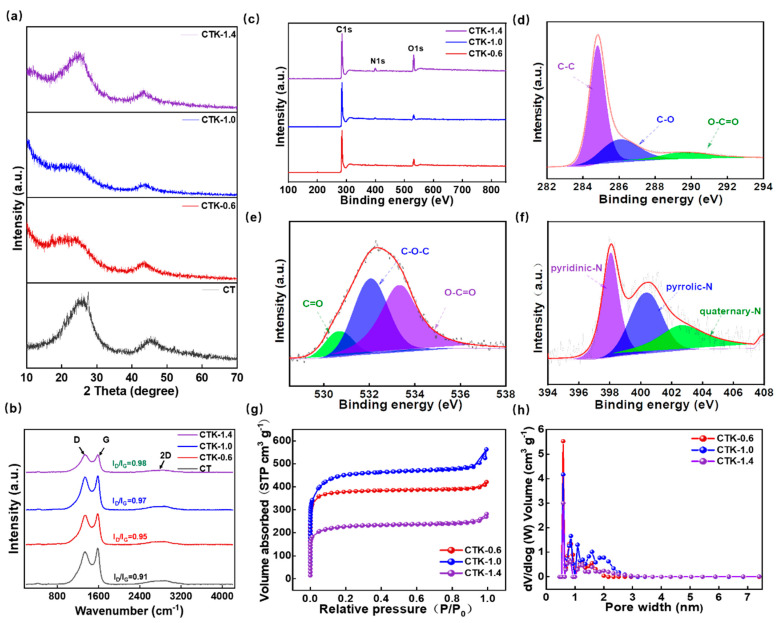
(**a**) XRD pattern, (**b**) Raman spectra, and (**c**) XPS spectra of prepared CNPCs with different g-C_3_N_4_ addition amounts. High-resolution XPS spectra of (**d**) C 1s, (**e**) O 1s, and (**f**) N 1s spectra of CTK-1.0. (**g**) N_2_ adsorption–desorption isotherm, and (**h**) pore size distribution curves of CNPCs.

**Figure 3 nanomaterials-13-02525-f003:**
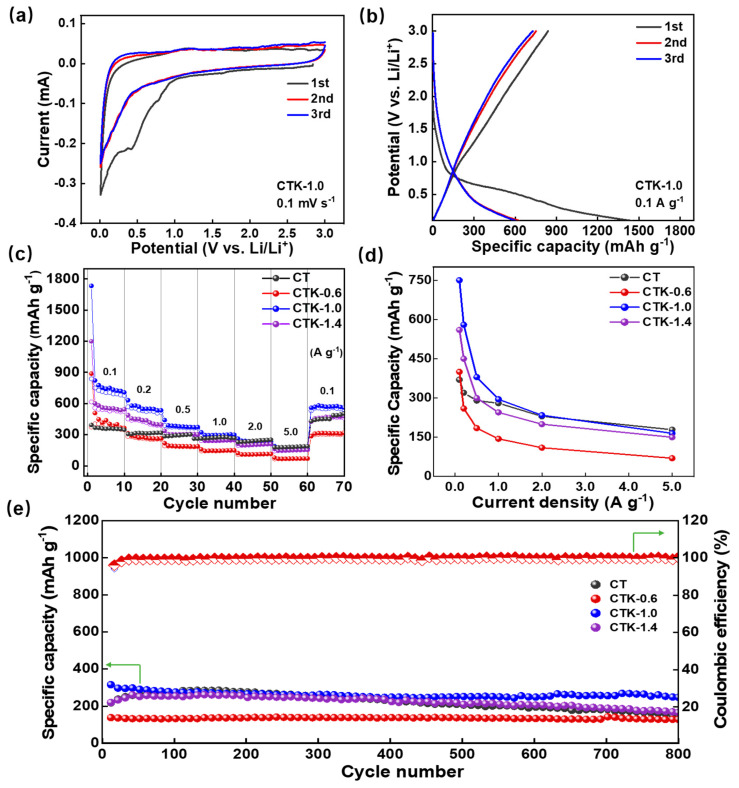
(**a**) CV curves and (**b**) constant charge–discharge curves of CTK-1.0. (**c**) The rate performance, (**d**) specific capacities at different current densities, and (**e**) cycling performance of prepared CT and CNPCs.

**Figure 4 nanomaterials-13-02525-f004:**
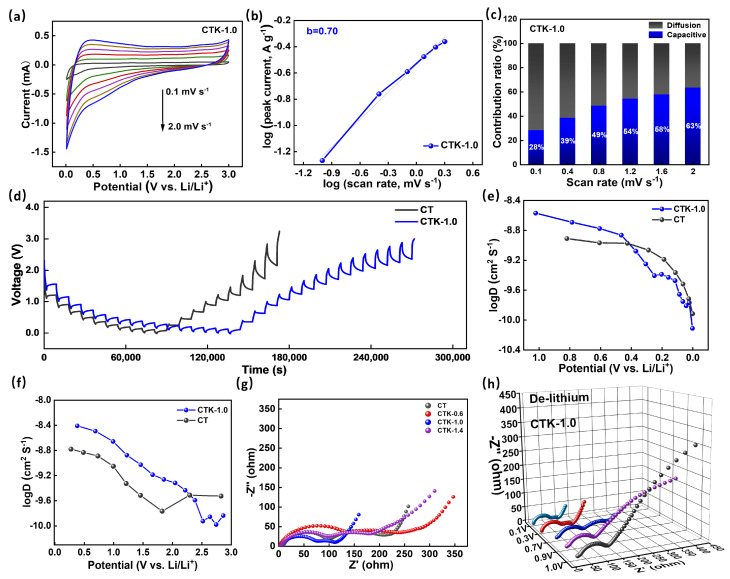
(**a**) The CV curves at different scanning rates, (**b**) the correlation of log (*i*) graph with log (*v*) graph, and (**c**) contribution ratios of the capacitive process at different scanning rates of CTK-1.0. (**d**) GITT curves of CTK-1.0 and CT. (**e**,**f**) The comparison of *D*_Li+_ calculated from GITT curves during the discharge and charge processes. (**g**) Nyquist plots of CNPCs and CT electrodes at 0.7 V. (**h**) Nyquist plots of CTK-1.0 electrodes at different operating voltages.

**Figure 5 nanomaterials-13-02525-f005:**
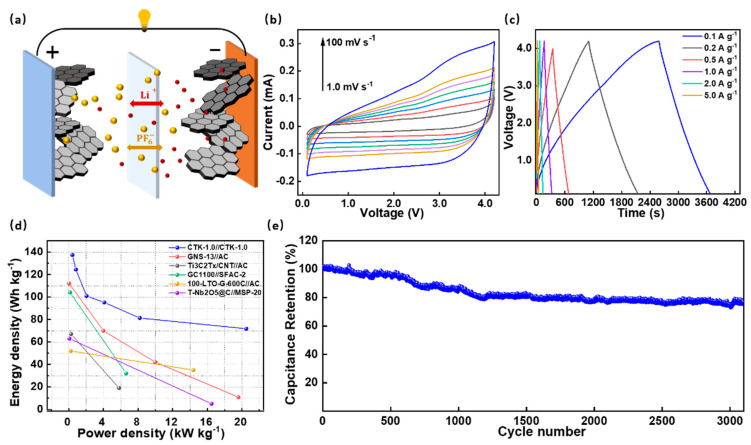
(**a**) Device construction diagram, (**b**) CV curves, (**c**) constant current charge–discharge profiles, (**d**) Ragone plots, and (**e**) cycling performance of the assembled DC-LICs (CTK-1.0//CTK-1.0).

## Data Availability

Data are contained within the article or Appendix A.

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
