# Peer review of "Nitrogen-Doped Porous Carbon Derived from Coal for High-Performance Dual-Carbon Lithium-Ion Capacitors"

_nanomaterials, 2023, doi:10.3390/nano13182525_

Round 1
Reviewer 1 Report
This paper is very well written, and I suggest publication. I have only minor comments for the authors to consider:
The doping of nitrogen in carbon materials has strong effects on their physical and chemical properties. The authors may be interested in earlier papers on this topic by Politzer et al: J. Mol. Model. 11, 1-7 (2005); Handbook of Semiconductor Nanostructures and Devices, Vol. 2, A. A. Balandin and K. I. King, eds., American Scientific Publishers, Los Angeles, 2006, ch. 7., and references cited therein.
Author Response
Reviewer #1 (Remarks to the Author):
This paper is very well written, and I suggest publication. I have only minor comments for the authors to consider:
Reply: We thank the reviewer for the encouraging comments and suggestions, which are valuable for us to improve the manuscript significantly. We hope that the reviewer finds the updated version of the manuscript to be suitable for its publication in Nanomaterials.
(1) The doping of nitrogen in carbon materials has strong effects on their physical and chemical properties. The authors may be interested in earlier papers on this topic by Politzer et al: J. Mol. Model. 11, 1-7 (2005); Handbook of Semiconductor Nanostructures and Devices, Vol. 2, A. A. Balandin and K. I. King, eds., American Scientific Publishers, Los Angeles, 2006, ch. 7., and references cited therein.
Reply: Thank the reviewer for the valuable suggestions. We have carefully studied the above-mentioned references, which are good references for our work. We have cited these references in the revised manuscript.
- Politzer, P.; Lane, P.; Murray, J.; Concha, M. Comparative analysis of surface electrostatic potentials of carbon, boron/nitrogen and carbon/boron/nitrogen model nanotubes. Journal of Molecular Modeling, 2005, 11, 1-7.
- Chen, W,; Joly, A.; Morgan, N.; Balandin, A.; Wang, K. Handbook of Semiconductor Nanostructures and Devices, vol 2. ch. 7. American Scientific Publishers, Los Angeles, 2006, 295-334.

Reviewer 2 Report
To authors
The results obtained are important in development of efficient materials for practical applications in energy storage technologies. The manuscript is easily readable concerning language, style and presentation. The references are appropriate and up to date. However, the authors asked to respond to the following minor comments:
1. Page 2. Replace “PH” to “pH”.
2. Page 2, 4. The labelling of the materials should be stated. For instance, K2CO4 or K2C2O4 (Fig. 1) is it the same material?

Author Response
Reviewer #2 (Remarks to the Author):
The results obtained are important in development of efficient materials for practical applications in energy storage technologies. The manuscript is easily readable concerning language, style and presentation. The references are appropriate and up to date. However, the authors asked to respond to the following minor comments:
Reply: We thank the reviewer for the encouraging comments and suggestions, which are valuable for us to improve the manuscript significantly. We hope that the reviewer finds the updated version of the manuscript to be suitable for its publication in Nanomaterials.
- 1. Page 2. Replace “PH” to “pH”.
Reply: We thank the reviewer for your insightful and constructive comments. We have replaced the “PH” to “pH” in the revised manuscript.
- Page 2, 4. The labelling of the materials should be stated. For instance, K2CO4 or K2C2O4 (Fig. 1) is it the same material?
Reply: Thank the reviewer for the valuable suggestions. The active agent used in this experiment is K2C2O4·H2O, thus the Fig1 is right, our description is wrong. In this regard, we have corrected this mistake, please see more details highlighted in the yellow background in the revised manuscript.

Reviewer 3 Report
This paper expained that Nitrogen-Doped Porous Carbon Derived from Coal for HighPerformance Dual-Carbon Lithium-ion Capacitors. This paper is well organized with logical consistency, and the quality of data analysis is satisfactory. However, some revision are required. Below is my comments.
1. Performance comparison with commercially available products is required.
2. XPS analysis of all samples is required. There will definitely be change of chemical states.
3. Electrochemical test evaluation followed by TEM or XPS analysis is required.
4. Correction of typos throughout the manuscript is required.
good
Author Response
Reviewer #3 (Remarks to the Author):
This paper expained that Nitrogen-Doped Porous Carbon Derived from Coal for High Performance Dual-Carbon Lithium-ion Capacitors. This paper is well organized with logical consistency, and the quality of data analysis is satisfactory. However, some revision are required. Below is my comments.
Reply: We thank the reviewer for the encouraging comments and suggestions, which are valuable for us to improve the manuscript significantly. We hope that the reviewer finds the updated version of the manuscript to be suitable for its publication in Nanomaterials.
- Performance comparison with commercially available products is required.
Reply: Thank you for your good suggestions. In this work, the assembled DC-LICs (CTK-1.0//CTK-1.0) delivers the maximum energy density of 137.6 Wh kg-1 (Based on the masses of the cathode and anode electrode) and a protracted lifetime of 3000 cycles. As a comparison, the energy density and power density of commercialized LICs are about 20~50 Wh kg-1 and ~30 kW kg-1, which is lower than that of our assembled devices. We have supplemented these descriptions of performance comparisons (Table S3), more details are highlighted in the yellow background in the revised manuscript.
- XPS analysis of all samples is required. There will definitely be change of chemical states.
Reply: We thank the reviewer for your constructive comments. The chemical states and element compositions of all samples have been investigated by XPS technology (Figure 2c), such as CTK-0.6, CTK-1.0, and CTK-1.4. The full XPS spectra of CNPCs show the presence of C, N, and O elements, and the detailed surface atomic contents have been summarized in Table S1. Moreover, as an illustration, the high-resolution XPS spectra of C 1s, O 1s, and N 1s have been fitted into different characteristic peaks (Figure 2d-f), which reflect the composition and valence states of the corresponding elements.
Figure 2. (c) XPS spectra of prepared CNPCs with different g-C3N4 addition amounts.
Table S1. Elements content obtained from the XPS data of all samples.
Sample |
C (at.%) |
N (at.%) |
O (at.%) |
S (at.%) |
CTK-0.6 |
90.85 |
1.26 |
7.78 |
0.10 |
CTK-1.0 |
91.74 |
3.03 |
5.02 |
0.11 |
CTK-1.4 |
82.41 |
4.34 |
13.12 |
0.13 |
- Electrochemical test evaluation followed by TEM or XPS analysis is required.
Reply: Thank you very much for your valuable suggestions. According to the results of the XRD pattern, the prepared coal-based nitrogen-doped porous carbon materials (CNPCs) are an amorphous structure. The HR-TEM image of CTK-1.0 is shown in Figure 1j, it can be found that the lattice fringes are not very clear due to its amorphous structure. In this regard, we have not performed TEM tests after the electrochemical performance measurements. Because not enough graphitization has been achieved, the lattice of the carbon material is not clear. In particular, we have systematically evaluated the electrochemical performances of CNPCs in this work, including CV, ZK, rate performance, and cycle stability, et, al. Moreover, the chemical states and element compositions of all samples have been investigated by XPS technology in revised manuscript (Figure 2c-f).
- Correction of typos throughout the manuscript is required.
Reply: Thank the reviewer for the valuable suggestions. We have checked the revised manuscript thoroughly and carefully to avoid potential spelling, grammar, and other mistakes. Please see more details highlighted in yellow background in the revised manuscript and Supporting Information.
